# Filtering-Assisted Airborne Point Cloud Semantic Segmentation for Transmission Lines

**DOI:** 10.3390/s24217028

**Published:** 2024-10-31

**Authors:** Wanjing Yan, Weifeng Ma, Xiaodong Wu, Chong Wang, Jianpeng Zhang, Yuncheng Deng

**Affiliations:** 1Faculty of Geography, Yunnan Normal University, Kunming 650500, China; 1943206000149@ynnu.edu.cn (W.Y.); 2133130005@ynnu.edu.cn (J.Z.); dengyckk@user.ynnu.edu.cn (Y.D.); 2Key Laboratory of Resources and Environmental Remote Sensing for Universities in Yunnan Kunming, Kunming 650500, China; 3Center for Geospatial Information Engineering and Technology of Yunnan Province, Kunming 650500, China; 4Power China Kunming Engineering Corporation Limited, Kunming 650051, China; wuxiaodong_kmy@powerchina.cn (X.W.); wc220350@163.com (C.W.); 5Engineering Research Center of 3D Real Scene, Kunming 650000, China

**Keywords:** airborne LiDAR, transmission line, semantic segmentation, cloth simulation filtering, machine learning

## Abstract

Point cloud semantic segmentation is crucial for identifying and analyzing transmission lines. Due to the number of point clouds being huge, complex scenes, and unbalanced sample proportion, the mainstream machine learning methods of point cloud segmentation cannot provide high efficiency and accuracy when extending to transmission line scenes. This paper proposes a filter-assisted airborne point cloud semantic segmentation for transmission lines. First, a large number of ground point clouds is identified by introducing the well-developed cloth simulation filter to alleviate the impact of the imbalance of the target object proportion on the classifier’s performance. The multi-dimensional features are then defined, and the classification model is trained to achieve the multi-element semantic segmentation of the transmission line scene. The experimental results and analysis indicate that the proposed filter-assisted algorithm can significantly improve the semantic segmentation performance of the transmission line point cloud, enhancing both the point cloud segmentation efficiency and accuracy by more than 25.46% and 3.15%, respectively. The filter-assisted point cloud semantic segmentation method reduces the volume of sample data, the number of sample classes, and the sample imbalance index in power line scenarios to a certain extent, thereby improving the classification accuracy of classifiers and reducing time consumption. This research holds significant theoretical reference value and engineering application potential for scene reconstruction and intelligent understanding of airborne laser point cloud transmission lines.

## 1. Introduction

As an important part of the power system [1], transmission lines are the focus of power network operation management and maintenance departments. Driven by national demand and sensor development, lightweight and small low-altitude flight platforms equipped with high-resolution sensors have been rapidly developed, including Manned or unmanned airborne LiDAR [2,3] (Light Detection and Ranging) measurement technology uses a non-contact, active measurement method to obtain high-density, high-precision three-dimensional space geometric and physical ancillary information of transmission lines (point cloud data) and realizes the description of spatial information with point as the smallest unit, offering advantages such as high efficiency, no line loss, and being unconstrained by regional conditions. It provides a new technical means for the detailed description and risk detection research of overhead transmission lines [4]. However, the data structure of airborne point clouds is discrete and lacks semantic information, so semantic segmentation can be performed before monomer modeling and scene simulation analysis.

Point cloud semantic segmentation refers to the classification of each point in a 3D point cloud into a predefined category. Currently, for the semantic segmentation of airborne point cloud data, scholars and research institutions both domestically and internationally have conducted numerous studies, and different point cloud semantic segmentation methods have been proposed. Rule-based segmentation methods [5,6,7] mostly use semantic segmentation rules preset by prior knowledge to categorize points into different semantic categories. However, these methods require complicated parameter adjustment and optimization, making them unsuitable for the segmentation of complex scenes such as pylons and large ground fluctuations and staggered lines. For the limitations of conventional rule-based approaches, some scholars use machine learning models [8,9,10,11] to predict the class probability of each point, including Random Forest, Support Vector Machine, Adaboost algorithm, etc. Guo et al. [12] calculated multi-dimensional classification features according to laser point cloud geometry and echo information, and used JointBoost classifier and context constraints to realize ground and transmission line classification. Toschi et al. [13] input five kinds of deep features into the random forest classifier to extract the transmission line point cloud and further realize the model reconstruction. Peng [14] et al. used Random Forest and Gradient Boosting Decision Tree to segment point clouds of different line scenes and analyze their reliability. In recent years, some scholars have applied deep learning algorithms [15,16,17] to point cloud semantic segmentation. However, due to the lack of architecture, the data representation is easy to lose, and there are problems such as occupying storage space and high requirements for calculation and memory, which are not suitable for large and complex scene classification. To address the sample imbalance problem in transmission line scenarios [18], Huang Jinglin et al. [19] used the BSMOTE algorithm to perform oversampling and synthetic processing on minority class samples located on the classification boundary, thereby improving the imbalance between samples. This method effectively enhances the model’s recognition capability for minority class samples, but the algorithm involves significant computational costs for identifying boundary samples and generating synthetic data, making it unsuitable for large-scale datasets. The document [20] proposes a hierarchical classification strategy for point clouds, filtering out non-ground points and classifying the LiDAR point clouds into ground points, non-ground points, noise points, and undefined points. However, this method requires manual segmentation of the undefined point clouds and is not suitable for areas with large volumes of point cloud data. Li et al. [21] proposed an improved adaptive surface interpolation method with a multilevel hierarchy by using cloth simulation and relief amplitude. This method achieves high filtering accuracy, but it is still a great challenge to accurately distinguish ground points in large complex areas with many outliers.

In summary, the problems associated with the semantic segmentation of point clouds based on transmission line scenes are mainly as follows: when the point cloud semantic segmentation model is applied to transmission line scenes, the complexity of point cloud data, uneven distribution, and large volume of data lead to low efficiency and insufficient accuracy when the point cloud segmentation algorithm is extended to transmission line scenes. To solve these issues, we propose a filter-assisted semantic segmentation method for airborne transmission line point clouds. Firstly, the ground points are filtered by cloth simulation filtering [22]. Then, a defined machine learning classifier model is employed to perform the semantic segmentation of transmission line scenes, and the transmission line is segmented into transmission lines, pylons, vegetation and ground, which lays a foundation for digital twin modeling and risk detection of transmission corridors.

The innovations of this paper include the following:(1)A point cloud semantic segmentation algorithm was designed that first separates the ground points, which constitute a larger proportion, to reduce sample imbalance and decrease the number of point clouds to be classified, thereby improving the accuracy and efficiency of point cloud classification in transmission line scenarios. This method effectively addresses the issue of being unable to simultaneously consider the accuracy and efficiency of point cloud classification in large-scale scenarios.(2)By comparing experimental data across different scenarios, this study analyzed the classification performance of commonly used machine learning classification algorithms such as AdaBoost (Ada), Random Forest (RF), K-Nearest Neighbors (KNN), Support Vector Machines (SVM), and Decision Trees (CART). It also utilized the Pearson correlation coefficient to analyze the correlation between influencing factors and the effectiveness of filter assistance. The strengths and weaknesses of different auxiliary algorithms were summarized, along with their applicability, providing technical support for detailed modeling of transmission line three-dimensional scenes.

The rest of this article is organized as follows: Section 2 presents the filtering-assisted point cloud semantic segmentation method and experimental materials; Section 3 analyzes the experimental results, including evaluations of semantic segmentation performed by different classifiers with filter assistance, the effectiveness of filter assistance under different experimental data, and a correlation analysis of influencing factors on the filter-assisted classification performance; Section 4 highlights the advantages and limitations of the proposed method; and Section 5 concludes this article.

## 2. Materials and Methods

### 2.1. Study Area

The airborne LiDAR system was employed to acquire the laser point cloud of a 22 KV high-voltage transmission line. Laser point cloud data acquisition equipment was DJI UAV M600 Pro equipped with a Rege VX-1LR lightweight and compact laser radar system with a maximum scanning point rate of 750,000 points/s, ranging accuracy of ±15 mm, and a designed flight height of 135 m. Data1 and Data3 are laser point clouds of transmission lines in the mountain area of Chaohu Hills, Anhui province, with geographical coordinates of 118°04′ E and 31°22′ N. The airborne LiDAR transmission line corridors are distributed with dense vegetation primarily comprising low shrubland, broad-leaved trees, and pine forests, as shown in Figure 1a,c. Data1 contains 4,243,654 point clouds with a total length of 1.9 km, and Data3 contains 6,478,324 point clouds with a total length of 1 km. Experimental Data2 consists of transmission line LiDAR point clouds from the Longyan plain area in Fujian Province, with geographic coordinates at 116°15′ E, 25°23′ N, as shown in Figure 1b. The transmission line corridor is distributed among natural land cover types such as farmland and woodland. It contains 3,704,276 point clouds, distributed in a ‘Y’-shape, with a total length of 0.8 km.

The transmission line is suspended between 10 and 60 m above the ground and contains five transmission lines, including two lightning protection lines and three conductive lines. The pylon comprises two categories: sheep-horn pylons and drum pylons. Power lines are separated from vegetation, while most pylon edges will be in contact with vegetation. The transmission line scenario suffers from an uneven distribution of point clouds and an imbalance of sample proportion, mainly manifested by a small number of point clouds for transmission lines and pylons but a large number for point clouds of vegetation and ground. Consisting of the point cloud data characteristics and the geometric differences of object types in the region, this paper divides the research data into four essential categories: vegetation, pylon, transmission line, and ground.

Considering the significant influence of sample category balance on machine learning point cloud segmentation [23], the point cloud data in the transmission line scenarios are unevenly distributed. The ground point cloud accounts for a large proportion and has more adjacency with other ground objects, leading to data imbalance and redundancy, which degrades the accuracy and efficiency of point cloud segmentation. This paper employs a filter-assisted approach to solve the above problems.

### 2.2. Methods

Given the low classification accuracy of point cloud data with unbalanced data proportions and large amounts of data using traditional machine learning methods, this paper utilizes cloth simulation to filter out the ground points with the largest number to reduce the influence of unbalanced data proportion and data redundancy on the accuracy and efficiency of point cloud segmentation. First, the laser point cloud data preprocessing software, CloudCompare, was employed to perform preprocessing tasks, such as cropping, denoising, and sampling, on the original point cloud data. Cloth simulation filtering is then performed to separate the ground and non-ground points. At the same time, the covariance matrix of the multi-scale point cloud was calculated based on the three-dimensional coordinate information of the point cloud, and the covariance matrix of each point was used to calculate the eigenvectors and eigenvalues, thereby obtaining 18 kinds of eigenvalues.

The feature set data was extracted from the labeled point clouds using the defined feature set. Seventy percent (70%) of the point clouds were randomly selected from the feature set data for training AdaBoost (Ada), Random Forest, K-Nearest Neighbor (KNN), Support Vector Machine (SVM), and Decision Tree (CART). The trained model predicts and classifies the remaining 30% of the point clouds. At the same time, the consumption time and accuracy of the classification results were evaluated to realize the semantic segmentation of the transmission line scene. Figure 2 shows the key steps of the proposed method.

#### 2.2.1. Point Cloud Data Filtering

The ground point filtering algorithm aims to take the lead in separating the ground points, with the most significant proportion, from the original point cloud data of the transmission line scene. This separation reduces the impact of sample proportion imbalance on the machine learning classifier’s performance, addresses the data redundancy phenomenon, and improves the classification efficiency. The cloth simulation filtering algorithm has been widely used because it fully utilizes the geometric and physical information of point cloud data and has a simple parameter setting without requiring too much prior knowledge. The realization process of the cloth simulation filtering algorithm includes cloth modeling and dynamic simulation. The virtual cloth comprises cloth nodes and springs between nodes. Additionally, the dynamic physical process simulation involves external forces and internal force constraints. The following equation shows the basic formula of the algorithm.
(1)$$m\frac{\partial^{2}X(t)}{\partial t^{2}}=F_{ext}(X,t)+F_{int}(X,t)$$
where m is the mass of the cloth particle, X is the position of the fabric node at time t, $F_{ext}(X,t)$ represents the external force factor (such as gravity and collision), and $F_{int}(X,t)$ represents the internal force constraint factor (the connection between nodes).

Cloth simulation filtering was applied to the transmission line point cloud data of the experimental data. Due to the presence of many low vegetation features in the transmission line scene of the test area, the cloth coefficient was set relatively hard to reduce the likelihood of objects being misidentified as ground objects. At the same time, considering the undulating terrain and the computer’s computational power, this paper performs cloth model parameter debugging.

The results show that for Data1, Data2, and Data3, when the cloth resolution is 0.7 m, the number of iterations is 500, the classification threshold is 0.5, and the cloth hardness coefficient is 1, the filtering effect of ground point clouds in the transmission line scene is optimal. For Data0, a cloth resolution of 0.3 m, 500 iterations, a classification threshold of 0.5, and a cloth hardness coefficient of 1 were used, resulting in a smaller number of ground point clouds compared to Data1. Figure 3 shows the filtered results of the experimental data. Table 1 shows the number of point clouds in each category and the total number of the experimental data before and after the filtering assistance.

#### 2.2.2. Non-Terrestrial Point Semantic Segmentation

Non-ground point cloud semantic segmentation mainly includes feature extraction, classifier model training, model evaluation, and model-based classification.

(1)Feature definition

Before semantic segmentation, sample features should be extracted according to the spatial distribution features of point clouds for the classifier to learn. Based on the coordinate data of the laser point cloud, the principal component analysis (PCA) algorithm was used in PyCharm2019.3.3 to calculate the covariance matrix of the point cloud in the neighborhood and solve the eigenvalues $\lambda_{1}$, $\lambda_{2}$, and $\lambda_{3}$, to obtain the dimensional features, reflecting the spatial distribution characteristics of the point set. The eigenvalues were sorted in descending order as $\lambda_{1}\geq \lambda_{2}\geq \lambda_{3}$.

The sample features are extracted according to the geometric characteristics of the point cloud of the transmission line scene. Since the transmission line is suspended in the air and keeps a specific distance from other ground objects, it also has a specific spatial density of thin cylindrical structures. Single transmission lines extend linearly and are parallel to each other [24], demonstrating the significance of the linearity, elevation, and density characteristics. The pylon point cloud has a metal frame structure with a wide bottom and a sharp top. It is distributed along the transmission line, and there is an interval between the pylons, allowing coordinate features, feature entropy, and sphericity to recognize it. The vegetation point cloud is located above the ground point cloud and has an irregular distribution, so it can be easily distinguished by feature entropy, elevation, and other features. The ground point cloud is located at the lowest data level and is evenly distributed with apparent planarity, anisotropy, elevation, and normal vector characteristics. Table 2 shows the spatially distributed characteristics of point clouds.

(2)Machine learning classifier

The semantic segmentation of the transmission line scene adopts the supervised learning method, which trains the segmented transmission line point cloud features while predicting the unsegmented point cloud using the trained model. AdaBoost (Adaptive Boosting, Ada), Random Forest (RF), K-Nearest Neighbors (KNN), Support Vector Machine (SVM), and Decision Tree (CART) were introduced to this classifier to verify the feasibility of filter-assisted machine learning point cloud segmentation. The machine learning algorithm inputs the sample features $X=[x_{1},x_{2},\ldots ,x_{n}]$ and sample labels $Y={\left[y_{1},y_{2},\ldots ,y_{n}\right]}^{T}$ into the classifier model $\bar{Y}=f$(X) to perform model learning. Moreover, the loss function is utilized to adjust the model to minimize the gap between the prediction result $\bar{Y}$ and Y to optimize the model. The trained classification model can be applied to the unknown dataset for prediction.

We considered the impact of parameter settings of machine learning classifiers on the semantic segmentation of point clouds based on power line scenarios. Due to the low classification accuracy under the default parameters of Adaboost and SVM algorithms, we optimized the parameters of these two algorithms. For the Adaboost algorithm, we used the SAMME (Stagewise Additive Modeling using a Multi-class Exponential error function) classification algorithm; the base classifier adopted decision tree classification; through a learning rate of 0.8, we controlled the contribution of the newly added weak classifiers in each iteration; the number of weak classifiers in the ensemble was 200, achieving an accuracy of 99.31%. The optimal parameter settings for the SVM algorithm used in this study were: the kernel function was RBF (Radial Basis Function), which is suitable for processing non-linear relationships and has the advantage of fewer parameters compared to polynomial kernels; the regularization parameter C was set to 0.9, controlling the balance between model complexity and overfitting; the parameter gamma, which controls the curvature of the decision boundary for the RBF kernel function, was set to ‘auto’, with a coefficient value of 1/n_features, where n_features is the number of features in the sample, achieving a maximum accuracy of 99.98%.

(3)Performance evaluation index

Accuracy, recall rate, and F1 score are the commonly used evaluation indicators [25]. This paper verifies the effect of filtering auxiliary methods on the classification performance of different classifiers based on accuracy and efficiency evaluation criteria. The overall accuracy evaluation index adopts precision, recall rate, and F1 score, and the efficiency evaluation index adopts the test time. The confusion matrix evaluates the classification accuracy of various point cloud data, obtaining the precision, recall, and F1 score. The target category of prediction is positive, and the other categories are negative. The classifier’s prediction on the test set is either correct or incorrect. The total results from these four scenarios are divided into TP (true positives), FP (false positives), TN (true negatives), and FN (false negatives).

Precision represents the proportion of actual positive samples in the predicted positive samples, expressed as follows:(2)$$P=\frac{TP}{TP+FP}$$

Recall rate is defined as the proportion of actual positive instances that are correctly identified as positive by the classifier, described as follows:(3)$$R=\frac{TP}{TP+FN}$$

F1 score is an evaluation index that considers both accuracy and recall rate [22]. Larger values for the F1 score indicate better performance. The F1 score is defined as follows:(4)$$F1=\frac{2TP}{2TP+FP+FN}$$

## 3. Experimental Results

The accuracy of the classification results before and after the filtering assistance was analyzed, and the accuracy, recall rate, and F1 score of the classifier, as well as the F1 score of each category, were counted as the accuracy evaluation index. The consumption time was employed as the efficiency evaluation index to analyze the experimental results.

### 3.1. Performance Evaluation of Different Classifiers in Transmission Line Scenarios

The proposed filter-assisted point cloud semantic segmentation method can improve the accuracy of each classifier and significantly reduce the consumption time. As shown in Table 3, the accuracy, recall, and F1 score of each classifier for the Data1 dataset were all below 96% before filtering. The accuracy of Ada, RF, KNN, SVM, and CART was improved by 4.15%, 7.43%, 6.95%, 4.89%, and 9.48%, respectively; the recall rate was increased by 4.38%, 7.42%, 6.99%, 6.17%, and 9.45%, respectively; and the F1 score was improved by 3.15%, 7.43%, 7.04%, 5.81%, and 9.47%, respectively. In addition, we also evaluated the time consumption before and after filtering of each classifier. After filtering assistance, the time consumption of Ada, RF, KNN, SVM and CART algorithms was reduced by 31.46%, 49.63%, 20.41%, 68.52% and 40.47%, respectively.

### 3.2. Filter-Assisted Evaluation of the Accuracy of Point Cloud Semantic Segmentation

By extracting features from the point cloud data of Data1 before and after cloth simulation filtering and performing machine learning point cloud semantic segmentation, the segmentation results of different algorithms are obtained, as shown in Figure 4, Figure 5 and Figure 6. For machine learning-based point cloud semantic segmentation, the classification accuracy of power lines is the highest, followed by pylons, vegetation, and ground points. It can be seen that the classification accuracy decreases with the increase in scene complexity and point cloud density in local areas. As shown in Figure 4, there is a significant difference between the vegetation point cloud before and after filtering assistance, and the accuracy of Ada, RF, KNN, SVM, and CART algorithms has been increased by 1.04%, 13.46%, 13.42%, 7.47%, and 16.71%, respectively. The AdaBoost algorithm partially misses some independent vegetation point clouds. After filtering, the overall number of missed vegetation point clouds is reduced, resulting in improved vegetation point cloud accuracy with the assistance of filtering. The leakage rate of point clouds decreases after vegetation filtering of the KNN algorithm. The random forest algorithm has a good effect on vegetation point cloud classification before and after filtering. Although the SVM algorithm misclassifies the point clouds of pylons and transmission lines into vegetation points, the classification effect is significantly improved after filtering assistance. In the classification of power lines, the accuracy of Ada, RF, KNN, SVM, and CART algorithms has been improved by 0.63%, 0.06%, 0.23%, 0.04%, and 0.06%, respectively. As shown in Figure 5, Ada, RF, KNN, SVM, and CART algorithms perform well in power line classification before and after filtering. The accuracy of the pylon category classification by the Ada, RF, KNN, SVM, and CART algorithms has been improved by 0.90%, 0.08%, 0.40%, 0.10%, and 0.32%, respectively. As shown in Figure 6, the KNN algorithm misclassifies the transmission lines near the pylons as pylon types. In contrast, Ada, RF, SVM, and CART algorithms provided a better classification effect.

### 3.3. Evaluation of the Accuracy of Filter-Assisted Point Cloud Semantic Segmentation in Different Transmission Line Scenarios

Considering the impact of terrain undulations and different ground point filtering effects on the method presented in this study, we set up two groups of verification data, Data2 and Data3, with different numbers of point clouds and degrees of terrain undulation. These were compared with Data1 to verify the point cloud classification performance under these conditions. At the same time, we set up Data0, which comes from the same data source as Data1 but has fewer ground points obtained through different filtering effects. This was used to validate the impact of the number of ground points on the filtering-assisted classification results through comparison.

(1)Time-consuming and precision evaluation based on different experimental scenarios

Figure 7 provides an evaluation of time consumption and accuracy based on different experimental scenarios. In terms of time consumption, the CSF + SVM algorithm has the highest time consumption, due to the significant time spent by the SVM in calculating the decision boundary. The CART algorithm has the shortest time consumption. Among the CSF + RF algorithm applications, Data1 took the longest time, while Data0 took the shortest time. In the CSF + ADA algorithm, the improvement effect of Data0 was the lowest. In terms of accuracy, within the Ada algorithm, Data3 showed the greatest improvement in accuracy after filtering assistance, followed by Data2, Data1, and Data0; in the RF algorithm, the improvement effect on accuracy was greatest for Data2, while Data1, Data3, and Data0 exhibited relatively high classification accuracy before and after filtering assistance. In the KNN algorithm, the improvement effect on accuracy was greatest for Data2, and Data1, Data3, and Data0 also showed relatively high classification accuracy before and after filtering assistance.

(2)Precision evaluation of element categories based on different experimental scenarios

We evaluated the specific class accuracy for different datasets, as illustrated in Figure 8. Among these, the CSF + KNN algorithm showed better time consumption improvement for Data3; the CSF + SVM algorithm and the CSF + CART algorithm performed better with Data1. The CSF + KNN algorithm had longer classification time consumption for the Data0 dataset; the reason for this may be that the KNN algorithm relies on distance measurement techniques to determine the similarity between data points. In the case of Data0, where ground point filtering is insufficient, the smaller the distance between ground point clouds and vegetation point clouds, the more similar their features are, requiring a large amount of computation for classification, thus leading to longer time consumption. On the whole, the transmission line scene has the best transmission line point cloud classification effect, followed by vegetation and tower point cloud, and filtering assistance has the best effect on improving the classification accuracy of vegetation point cloud.

(3)Correlation factor analysis of the classification efficiency of different algorithms assisted by filtering

Considering that factors such as terrain undulation, the number of point clouds, class imbalance, and the number of classes may all have an impact on point cloud classification results, we conducted a correlation analysis between the F1 scores obtained from classification with different experimental data and these influencing factors. Terrain undulation is represented by the maximum elevation difference of scene ground points (the maximum elevation value of ground points minus the minimum elevation value of ground points); the class imbalance index is expressed as (the maximum number of points in a class minus the minimum number of points in a class) divided by the maximum number of points in a class. The calculation results of each influencing factor are shown in Table 4. By introducing the Pearson correlation coefficient (denoted as r) to measure the degree of linear correlation between the accuracy of experimental data and influencing factors, the value of the Pearson correlation coefficient ranges from −1 to 1, where 1 indicates perfect positive correlation, −1 indicates perfect negative correlation, and 0 indicates no linear correlation. Pearson correlation coefficients are categorized as follows: a range of 0–0.3 indicates a weak correlation; a range of 0.3–0.7 is considered moderate correlation; a range of 0.7–1 indicates a strong correlation. Table 5 shows the correlations of accuracy improvements and time consumption.

Table 4 shows the correlation of accuracy with factors for different classification algorithms. Among these, the improvement in accuracy of the Ada algorithm after filtering exhibits a strong negative correlation with the sample imbalance index and the number of classes, a moderate negative correlation with terrain undulation, and a strong positive correlation with the number of point clouds. This indicates that the higher the balance index and the number of sample classes, along with greater terrain undulation, the lower the classification accuracy of the Ada algorithm; however, a larger sample size is likely to increase the training accuracy. The classification accuracy of the RF algorithm shows a moderate negative correlation with terrain undulation; the greater the terrain undulation, the more likely it is to reduce its classification accuracy to some extent. The classification accuracy of the KNN algorithm shows a moderate negative correlation with the sample imbalance index and the number of sample classes; the SVM algorithm’s classification accuracy shows a moderate negative correlation with the number of sample classes and a weak negative correlation with the sample imbalance index. The classification accuracy of the CART algorithm shows a strong negative correlation with the sample imbalance index and a weak negative correlation with the number of sample classes.

Table 6 shows the correlation of time consumption with factors for different classification algorithms. Among these, the time consumption of the Ada algorithm shows a strong positive correlation with the sample size, sample imbalance index, and terrain undulation, and a moderate correlation with the number of classes; the RF algorithm shows a moderate positive correlation with the sample size, sample imbalance index, terrain undulation, and the number of classes; the time consumption of the KNN algorithm is minimal and is largely unaffected by these factors; the time consumption of the SVM algorithm shows a moderate positive correlation with sample imbalance and sample size, meaning the larger the sample imbalance and sample size, the longer the time consumption. The time consumption of the CART algorithm shows a strong positive correlation with sample imbalance and sample size, and a moderate positive correlation with terrain undulation and the number of classes.

Overall, the filter-assisted point cloud semantic segmentation method tends to reduce the volume of sample data, the number of sample classes, and the sample imbalance index. Through correlation analysis, it was found that the accuracy and time consumption of each algorithm are affected to varying degrees by the sample size, sample imbalance index, and the number of sample classes. This influence tends to result in improved classification accuracy and reduced time consumption when assisted by filtering, indicating that filter-assisted algorithms effectively enhance the efficiency of point cloud semantic segmentation in power line scenarios. Additionally, we found that the accuracy of the Ada algorithm and RF is still influenced to a certain extent by terrain undulation, whereas the classification accuracy of the KNN, SVM, and CART algorithms is hardly affected by terrain undulation. The KNN algorithm has very low time consumption, which is hardly affected by these factors. However, the KNN algorithm does require longer computation times when dealing with point clouds of different classes that are very close to each other.

## 4. Discussion

Experiments demonstrate the proposed method’s superiority over other existing single machine learning methods. Moreover, the experimental data selected in this study generally represent power grid engineering, extending the application scope of the proposed method. The superiority of this method can be attributed to the following aspects:

In the point cloud data pre-processing stage, the ground point cloud with the largest area and number is filtered through the cloth simulation filtering algorithm based on the original data, significantly reducing the number of point clouds in the transmission line scene while effectively retaining the transmission line, pylon, and vegetation point clouds. This addresses the impact of sample proportion imbalance on the point cloud segmentation accuracy. During the point cloud semantic segmentation stage, 18 features are created according to the spatial structure distribution of the transmission line scene to ensure the completeness and accuracy of the semantic segmentation results of the transmission line field point cloud.

The filter-assisted point cloud semantic segmentation method will reduce the sample data size, sample category number and sample imbalance index of the transmission line scenario to some extent, thus improving the classification accuracy of the classifier and reducing the time consumption. The accuracy of the Ada algorithm and RF is also affected by surface fluctuation to some extent, and the classification accuracy of KNN, SVM and CART algorithms is almost unaffected by surface fluctuation. In the transmission line scenario, the transmission line point cloud classification effect is the best, followed by the vegetation and the tower point cloud, and the filter assistance has the best effect on improving the classification accuracy of the vegetation point cloud. At the same time, compared with filtering the point cloud data with fewer ground points, the data with sufficient ground point filtering has higher classification accuracy. However, the time-consuming KNN algorithm is more sensitive to insufficient ground point filtering and takes a long operation time. The AdaBoost algorithm, combined with a weak classifier, improves the accuracy of the whole model, is robust to noise and outliers, and achieves the best overall classification effect. The RF algorithm integrates the prediction results of multiple decision trees to provide high prediction and classification accuracy. Meanwhile, its highly parallelized feature makes it suitable for processing large-scale datasets. The definition of decision boundary affects the classification effect of the SVM algorithm, so the classification accuracy is low when there are more categories, and the classification accuracy is improved after the ground point at the filter. Although the CART algorithm uses a tree structure to recursively segment data and capture the nonlinear relationship within the data, overfitting the training data leads to poor performance and low classification accuracy on new datasets. After the filtering assistance, the consumption time of Ada, RF, KNN, SVM, and CART algorithms was reduced by 31.46%, 49.63%, 20.41%, 68.52%, and 40.47%, respectively. Among them, the CART algorithm preferentially selects features with large values to train the model, requiring a lower running time. The KNN algorithm requires much time to calculate the distance value for each data point in the dataset with a large amount of data. The filtering improves the training efficiency. Although the RF employs the bagging integration algorithm, which can parallelize calculations, it still requires a relatively long running time for high-dimensional data. AdaBoost adopts the boosting integration algorithm. Although AdaBoost should run the serialization method, it only needs fewer computing resources and time to train the model because it utilizes a single base classifier. SVM takes the longest time. When the transmission line scenario is large and the number of sample point clouds is too much, the filter-assisted SVM algorithm is still faced with the problem that the decision surface calculation takes too long and is not applicable, which can be improved by the help of downsampling operations.

Generally, machine learning algorithms are affected by unbalanced point cloud data samples, a large amount of data, and complex scenes in transmission line scenarios to varying degrees. The cloth simulation filtering method can effectively reduce the influence of these factors and improve the accuracy and efficiency of semantic segmentation of machine learning algorithms. The proposed filter-assisted machine semantic segmentation method provides higher accuracy than the method using a single machine learning classifier. Accordingly, the segmentation advantages of power lines and pylon categories are evident, and the filtering assistance can significantly enhance the point cloud segmentation of vegetation categories.

## 5. Conclusions

Airborne LiDAR point cloud semantic segmentation is crucial in point cloud data processing. Due to the large volume of point clouds in transmission line scenarios, complex scenes, and unbalanced sample proportions, mainstream machine learning-based point cloud segmentation algorithms exhibit low efficiency and insufficient precision when applied to transmission line scenarios. This paper proposed a filter-assisted point cloud semantic segmentation method based on machine learning, which can simplify data processing and improve the speed and efficiency of model classification. First, the original point cloud data is filtered by cloth simulation. Then, the machine learning classifier model performs the semantic segmentation. The point cloud classification accuracy of each classifier assisted by filtering is higher than that of single machine learning-based classification methods. Moreover, the accuracy, recall rate, and F1 index are increased by 4.15%, 4.38%, and 3.15%, respectively, while the time consumption is reduced by 25.46% or more.

Although the point cloud semantic segmentation method with the introduction of the cloth simulation filtering algorithm assisted by machine learning improves the accuracy and efficiency, the following future research aspects should be further studied: First, the computational complexity of the algorithm is high, which requires a lot of computing resources and time. Therefore, the algorithm’s parameter setting should be optimized, and the degree of automation of the filtering algorithm should be improved. Second, the machine learning algorithm is limited by subjective feature definition and other factors, limiting its application in the point cloud segmentation of large-scale transmission lines to a certain extent. Therefore, the filter-assisted classification should be extended to the deep learning point cloud classification algorithms, and the research on the fine separation of transmission lines should be further pursued.

## Figures and Tables

**Figure 1 sensors-24-07028-f001:**
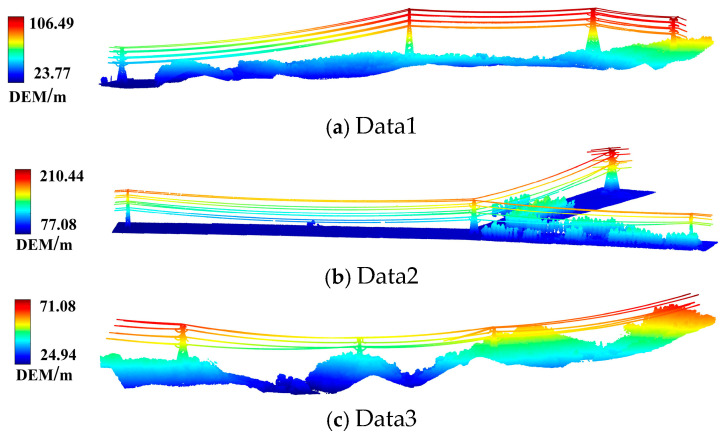
Experimental data.

**Figure 2 sensors-24-07028-f002:**
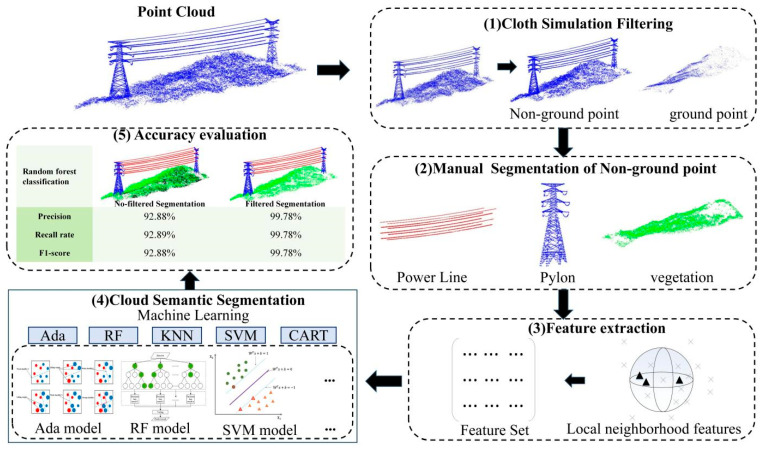
Key steps of the proposed method (the black triangle in (3) represents the point cloud of the best field, and the different colors and triangles in (4) represents the point cloud identified as a certain category).

**Figure 3 sensors-24-07028-f003:**
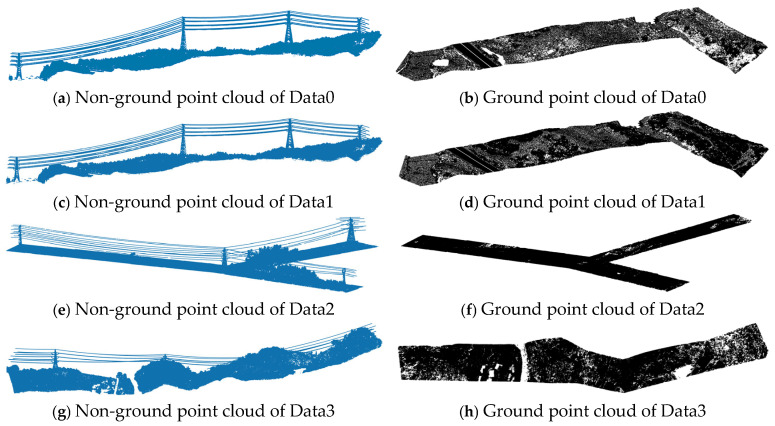
Filtered ground and non-ground points of the experimental data.

**Figure 4 sensors-24-07028-f004:**
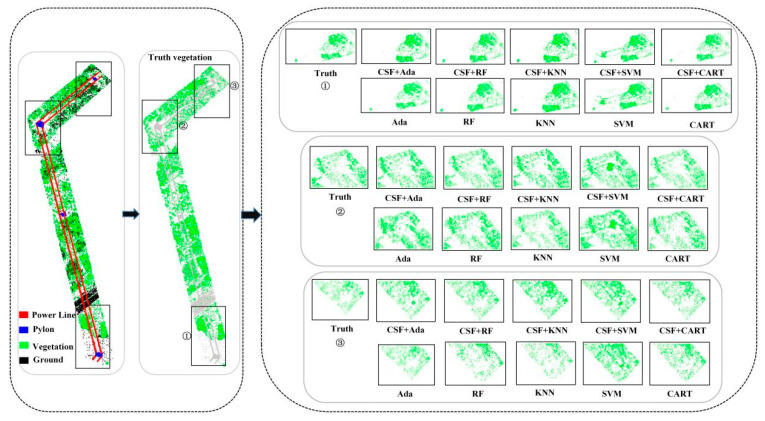
Classification results of vegetation before and after filter assistance by different machine learning methods based on Data1: ①, ② and ③, respectively, indicate the vegetation point clouds of two areas in the scene.

**Figure 5 sensors-24-07028-f005:**
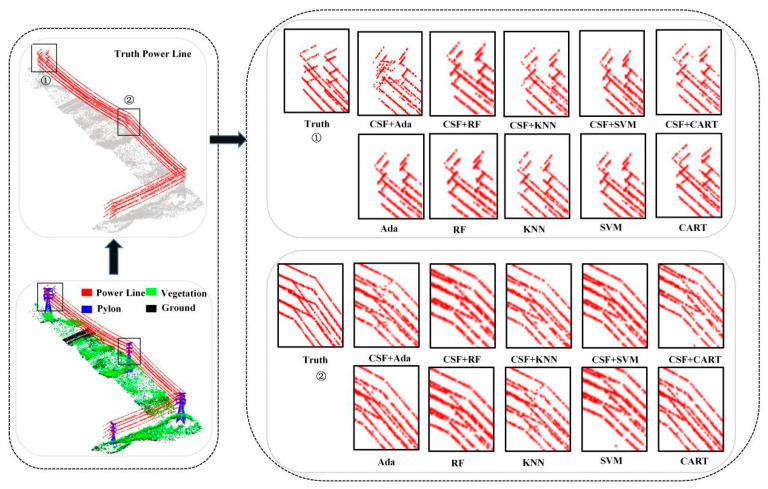
Classification results of power lines before and after filter assistance by different machine learning methods: ① and ②, respectively, indicate the transmission line point clouds of two areas in the scene.

**Figure 6 sensors-24-07028-f006:**
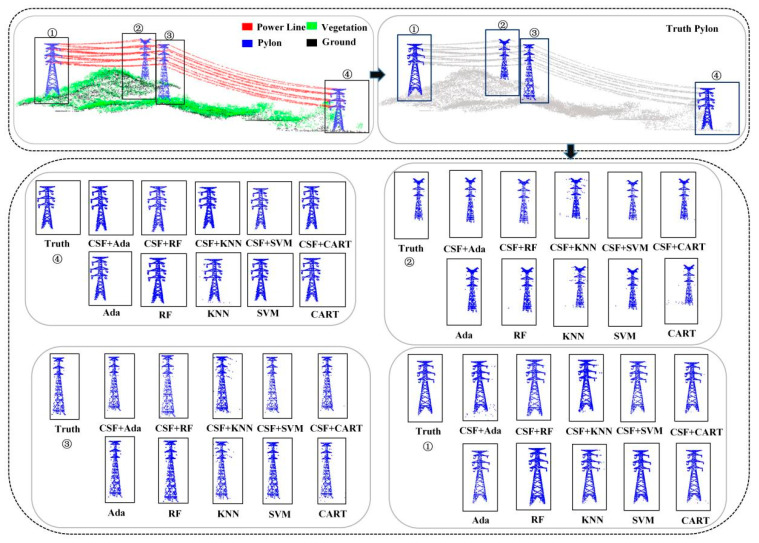
Classification results of different machine learning methods before and after filtering: ①, ②, ③, and ④ indicate the numbers of the four pylons in the scene, respectively.

**Figure 7 sensors-24-07028-f007:**
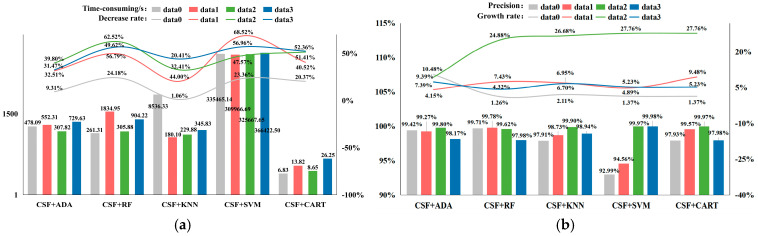
Evaluation of the efficiency based on the different classifiers: (**a**) time-consuming and reduction rates comparison of the four groups of experiments; (**b**) overall accuracy and growth rates comparison of the four experiments.

**Figure 8 sensors-24-07028-f008:**
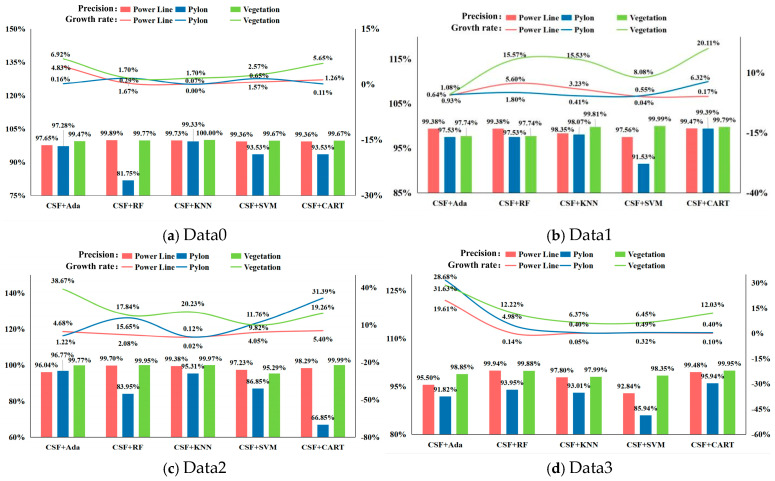
Evaluation of site classification accuracy based on different experimental data: (**a**–**d**) indicate the precision and precision growth rate of power line, pylon, and vegetation in Data0, Data1, Data2, and Data3.

**Table 1 sensors-24-07028-t001:** Number of point cloud categories for the experimental data.

	Filtering	Data0	Data1	Data2	Data3
Power Line	-	66,142	66,142	35,494	201,837
Pylon		49,892	49,891	39,376	86,680
Vegetation		3,805,657	3,290,413	2,643,882	4,181,197
Ground		321,865	837,110	987,718	2,150,438
Total quantity		4,243,556	4,243,556	3,706,471	6,620,152
Power Line	CSF	66,142	66,142	35,494	201,837
Pylon		49,892	49,891	39,376	86,680
Vegetation		3,805,657	3,290,413	2,643,882	4,181,197
Ground		-			
Total quantity		3,921,691	3,406,446	2,718,752	4,469,714

**Table 2 sensors-24-07028-t002:** Spatially distributed characteristics of point clouds.

Feature Type	Formula	Feature Type	Formula
Linear	$L_{\lambda }=\frac{\lambda_{1}-\lambda_{2}}{\lambda_{1}}$	Sphericity	$S_{\lambda }=\frac{\lambda_{3}}{\lambda_{1}}$
Planarity	$P_{\lambda }=\frac{\lambda_{2}-\lambda_{3}}{\lambda_{1}}$	Curvature Change	$C_{\lambda }=\frac{\lambda_{3}}{\lambda_{1}+\lambda_{2}+\lambda_{3}}$
Anisotropy	$A_{\lambda }=\frac{\lambda_{1}-\lambda_{3}}{\lambda_{1}}$	Isotropy	$O_{\lambda }=\sqrt[{3}]{\lambda_{1}+\lambda_{2}+\lambda_{3}}$
Term Entropy	$E_{\lambda }=-\sum_{i=1}^{3}\lambda_{i}\ln\left(\lambda_{i}\right)$	Sum of Eigenvalues	$R_{\lambda }=\lambda_{1}+\lambda_{2}+\lambda_{3}$
Angle between principal direction and normal vector (θ)	$\cos\theta =(a\cdot b)/(\left|a\right|+\left|b\right|)$	GPS Time	GPS_Time
XYZ Coordinate	X,Y,Z	Eigenvalue	$\lambda_{1},\lambda_{2},\lambda_{3}$
Echo Intensity	Instenst	Echo frequency	Return Number

**Table 3 sensors-24-07028-t003:** Evaluation of the accuracy and efficiency of the filter-assisted pre-classifier based on Data1.

Parameter	Ada	CSF+ Ada	RF	CSF+ RF	KNN	CSF+ KNN	SVM	CSF+ SVM	CART	CSF+ CART
Precision	95.31%	99.27%	92.88%	99.78%	92.31%	98.73%	90.15%	94.56%	90.95%	99.57%
Recall rate	95.35%	99.53%	92.89%	99.78%	92.32%	98.77%	87.56%	92.96%	90.97%	99.57%
F1-score	95.32%	98.32%	92.88%	99.78%	92.25%	98.74%	88.25%	93.38%	90.96%	99.57%
Time-consuming/s	533.89	365.89	1506.63	758.98	155.54	123.80	984,775.10	309,966.69	8.86	5.27

**Table 4 sensors-24-07028-t004:** Calculation results of various correlating factors.

	Filtering	Category Imbalance Index	Sample Size	Terrain Ruggedness	Sample Class Number
Data0	-	1,732,536.88	4,243,556	54.58	4
Data1		1,736,607.03	4,243,556	54.58	4
Data2		1,522,047.14	3,706,471	7.14	4
Data3		2,741,756.60	6,620,152	104.06	4
Data0	CSF	1,603,876.95	3,921,691	54.58	3
Data1		1,392,797.02	3,406,446	54.58	3
Data2		1,112,041.48	2,718,752	7.14	3
Data3		1,825,320.60	4,469,714	104.06	3

**Table 5 sensors-24-07028-t005:** Correlation of accuracy with factors for different classification algorithms.

	Ada	RF	KNN	SVM	CART
Category imbalance index	−0.70	0.08	−0.62	−0.43	−0.79
sample size	0.83	0.07	−0.04	0.05	0.14
terrain ruggedness	−0.32	−0.37	0.08	−0.04	0.13
Sample class number	−0.90	0.04	−0.67	−0.76	−0.25

**Table 6 sensors-24-07028-t006:** Correlation of time consumption with factors for different classification algorithms.

	Ada	RF	KNN	SVM	CART
Category imbalance index	0.92	0.37	−0.01	0.59	0.89
sample size	0.92	0.37	−0.01	0.59	0.89
terrain ruggedness	0.78	0.30	0.01	−0.02	0.66
Sample number	0.48	0.41	0.02	0.20	0.41

## Data Availability

The data that support the findings of this study are available on request from the corresponding author (W.M.), upon reasonable request.

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
