# Peer review of "Filtering-Assisted Airborne Point Cloud Semantic Segmentation for Transmission Lines"

_sensors, 2024, doi:10.3390/s24217028_

Round 1
Reviewer 1 Report
Comments and Suggestions for Authors
1. Please provide a more detailed explanation of the main work content.
2. Please elaborate on the data sample size (number of categories, total quantity for each category), explain how ground truth was obtained, and describe how the training and test sets were divided.
3. Please provide a comparison of the data volume before and after the original point cloud filtering.
4. Optimize the visualization results to highlight the superiority of the method presented in this paper.
5. Please discuss the limitations of the proposed method.
6. The cited literature is rather outdated. A more in-depth literature review should be provided, especially regarding state-of-the-art methods for this task.
Author Response
Response to Reviewers’ Comments
Remote sensing
Manuscript Number: sensors-3211209
Manuscript Title: Filtering-assisted airborne point cloud semantic segmentation for transmission lines
General Comments: We are extremely grateful to the esteemed editors and reviewers for reviewing my work and providing detailed comments and suggestions for revision. I really appreciate Editor giving me a chance to revise. Thanks a lot. I have revised the original letter according to reviewers, including revising abstract and introduction, increasing validation experiments,enhancing the discussion section,adding explanations, polishing the letter and so on. A revision location is marked in the letter with a notation like (Revised edition: P1, Line 12-36), which means that the revision occurs on page 1, from Line 12 in the left side to the Line 36 in the right side of the document .
A list of changes and responses to reviewers are as follows. A copy of the fully revised manuscript that has all the changes highlighted in red color(Revised edition_highlight in red), along with grammatical corrections(Revised edition_remarked) is uploaded as an attachment to the email as well. Appreciate for your consideration.
Reviewer #1:
Comment and Suggestions for Authors:
The paper describes the methodology to estimate forest biomass by combining LiDAR data and features from optical remote sensing data at high spatial resolution. Empirical allometric model is used to compute biomass.
- Please provide a more detailed explanation of the main work content.
Author Response:
I would like to express my gratitude for the useful comments on the letter. We have provided a more detailed explanation of the main work content in a revised version.. (Revised edition: P7, Line 116-122)
- Please elaborate on the data sample size (number of categories, total quantity for each category), explain how ground truth was obtained, and describe how the training and test sets were divided.
Author Response:
The insightful and helpful suggestions are greatly appreciated. Our initial letter did not clearly describe the information on the data samples, as shown in Table 1, and we have addressed these issues in a revised version.The “ground truth” we used is the ground point obtained by the cloth simulation filter algorithm, which is used as the correct sample label when performing machine learning classification, and is also used to verify the test classification accuracy. At the same time, We added three sets of experimental data on the basis of the original, and divided the data amount into the training set and the test set by 7:3 for machine learning classification.(Revised edition: P6, Line 236)
Table 1. Number of point cloud categories for the experimental data
|
Filtering |
Data0 |
Data1 |
Data2 |
Data3 |
Power Line |
- |
66142 |
66142 |
35494 |
201837 |
Pylon |
49892 |
49891 |
39376 |
86680 |
|
Vegetation |
3805657 |
3290413 |
2643882 |
4181197 |
|
Ground |
321865 |
837110 |
987718 |
2150438 |
|
Total quantity |
4243556 |
4243556 |
3706471 |
6620152 |
|
Power Line |
CSF |
66142 |
66142 |
35494 |
201837 |
Pylon |
49892 |
49891 |
39376 |
86680 |
|
Vegetation |
3805657 |
3290413 |
2643882 |
4181197 |
|
Ground |
- |
||||
Total quantity |
3921691 |
3406446 |
2718752 |
4469714 |
- Please provide a comparison of the data volume before and after the original point cloud filtering.
Author Response:
The insightful and helpful suggestions are greatly appreciated. Our original letter did not clearly describe the comparison of data volumes before and after the original point cloud filtering, and we have addressed these issues in a revised version,shown in Table 1.(Revised edition: P6, Line 236)
- Optimize the visualization results to highlight the superiority of the method presented in this paper.
Author Response:
Thank you for the valuable comments and we extend our sincerest gratitude to reviewer. We optimized the visualization results to highlight the superiority of the method described here, as shown in Figure 1 (Revised edition: P4, Line 141), Figure 3 (Revised edition: P6, Line 235), Figure 7 (Revised edition: P11, Line 388), Figure 8 (Revised edition: P12, Line 418).
- Please discuss the limitations of the proposed method.
Author Response:
Thank you for the valuable comments and we extend our sincerest gratitude to reviewer. We have added the discussion of the limitations of the proposed direction to the discussion section of the revision.(Revised edition: P16, Line 547-550)
- The cited literature is rather outdated. A more in-depth literature review should be provided, especially regarding state-of-the-art methods for this task.
Author Response:
Thank you for the valuable comments and we extend our sincerest gratitude to reviewer. The literature cited in our original letter is more outdated, and we have introduced state-of-the-art methods in the introduction section of the revised version to modify the literature review. (Revised edition: P2, Line 65-88)
All in all, thank you very much for your reconsidering our revised manuscript for potential publication in Sensors. I'm looking forward to hearing from you soon. Correspondence should be addressed to Wanjing Yan or Weifeng Ma at the following address.

Reviewer 2 Report
Comments and Suggestions for Authors
According to this manuscript's contribution part, it has 2 contribution point:
(1).A point cloud semantic segmentation algorithm assisted by cloth simulation filtering is designed to improve the accuracy and efficiency of transmission line point cloud classification by first separating the ground points with the largest proportion.
(2).the performance of common machine learning classification algorithms such as Adaboost, random forest algorithm, k-nearest neighbor algorithm, support vector machine and decision tree is compared and analyzed, and the advantages and disadvantages of different classifiers are summarized, which provides technical support for the fine modeling of 3D scenes of transmission lines.
To me, 1st point, there are many ways to find similary content,
An Adaptive Surface Interpolation Filter Using Cloth Simulation and Relief Amplitude for Airborne Laser Scanning Data.Remote Sens. 2021, 13(15), 2938; https://doi.org/10.3390/rs13152938 ;
https://blog.csdn.net/gitblog_00565/article/details/141006718.
2nd point, on the basis of the first innovation point, reproducing multiple classic algorithms, analyzing and comparing their performance is a good experiment, but it cannot be considered innovative.
So, I think the article lacks innovation and does not meet the level of this journal.
In other aspects, such as writing and image matching, I think this article is still good.
In terms of references, the format is not consistent, and the overall literature tends to be outdated.
I think this paper overall conforms to the format of the paper, and the experiments look good, but in terms of innovation, it is not suitable for publication in “Sensors” (Q1, JCR2) (my standard is at least 2 strong innovation points). It is recommended to choose other journals.
Author Response
Response to Reviewers’ Comments
Remote sensing
Manuscript Number: sensors-3211209
Manuscript Title: Filtering-assisted airborne point cloud semantic segmentation for transmission lines
General Comments: We are extremely grateful to the esteemed editors and reviewers for reviewing my work and providing detailed comments and suggestions for revision. I really appreciate Editor giving me a chance to revise. Thanks a lot. I have revised the original letter according to reviewers, including revising abstract and introduction, increasing validation experiments,enhancing the discussion section,adding explanations, polishing the letter and so on. A revision location is marked in the letter with a notation like (Revised edition: P1, Line 12-36), which means that the revision occurs on page 1, from Line 12 in the left side to the Line 36 in the right side of the document .
A list of changes and responses to reviewers are as follows. A copy of the fully revised manuscript that has all the changes highlighted in red color(Revised edition_highlight in red), along with grammatical corrections(Revised edition_remarked) is uploaded as an attachment to the email as well. Appreciate for your consideration.
Reviewer #2:
Comment and Suggestions for Authors:
According to this manuscript's contribution part, it has 2 contribution point:
(1).A point cloud semantic segmentation algorithm assisted by cloth simulation filtering is designed to improve the accuracy and efficiency of transmission line point cloud classification by first separating the ground points with the largest proportion.
(2).the performance of common machine learning classification algorithms such as Adaboost, random forest algorithm, k-nearest neighbor algorithm, support vector machine and decision tree is compared and analyzed, and the advantages and disadvantages of different classifiers are summarized, which provides technical support for the fine modeling of 3D scenes of transmission lines.
To me, 1st point, there are many ways to find similary content,
An Adaptive Surface Interpolation Filter Using Cloth Simulation and Relief Amplitude for Airborne Laser Scanning Data.Remote Sens. 2021, 13(15), 2938; https://doi.org/10.3390/rs13152938 ;
https://blog.csdn.net/gitblog_00565/article/details/141006718.
Author Response:
The insightful and helpful suggestions are greatly appreciated. I have read the article, we consider it important to the writing of the manuscript and quote the article during the revision process. In the previous manuscript, I did not consider the impact of the terrain fluctuation and the change of the number of point clouds during the experiment. We added the verification experiment to enhance the persuasion of our proposed filter assistance method. Li et al. proposed an improved adaptive surface interpolation method with a multilevel hierarchy by using cloth simulation and relief amplitude. This method achieves high filtering accuracy, but it is still a great challenge to accurately distinguish ground points in large complex areas with many outliers. (Revised edition: P2, Line 84-88).
2nd point, on the basis of the first innovation point, reproducing multiple classic algorithms, analyzing and comparing their performance is a good experiment, but it cannot be considered innovative.
So, I think the article lacks innovation and does not meet the level of this journal.
Author Response:
We appreciate your valuable suggestions regarding our manuscript and your interest in previously published research in the field. In response to your comments, we would like to elucidate how our study contributes to existing knowledge and advances the field.
Firstly, our revised innovations are as follows:
(1)A point cloud semantic segmentation algorithm was designed that first separates the ground points, which constitute a larger proportion, to reduce sample imbalance and decrease the number of point clouds to be classified, thereby improving the accuracy and efficiency of point cloud classification in transmission line scenarios. This method effectively addresses the issue of being unable to simultaneously consider the accuracy and efficiency of point cloud classification in large-scale scenarios.
(2)By comparing experimental data across different scenarios, this study analyzed the classification performance of commonly used machine learning classification algorithms such as AdaBoost(Ada), Random Forest(RF), K-Nearest Neighbors (KNN), Support Vector Machines (SVM), and Decision Trees(CART). It also utilized the Pearson correlation coefficient to analyze the correlation between influencing factors and the effectiveness of filter assistance. The strengths and weaknesses of different auxiliary algorithms were summarized, along with their applicability, providing technical support for detailed modeling of transmission line three-dimensional scenes.
We revised it on the basis of the original statement (Revised edition: P3, Line 102-115).
Secondly, we added three sets of comparison verification data to validate the semantic segmentation performance of the method presented in this paper under different ground point filtering effects and varying topographic conditions in transmission line scenarios. Additionally, considering factors such as terrain undulation degree, number of point clouds, class imbalance, and number of categories, we will conduct a correlation analysis between the results from different classification algorithms and these influencing factors. This will enhance the persuasiveness of our method and also allow us to compare and analyze the applicability of different classification methods (Revised edition: P11, Line 368-498).
Finally, the insightful and helpful suggestions are greatly appreciated.
In other aspects, such as writing and image matching, I think this article is still good.
In terms of references, the format is not consistent, and the overall literature tends to be outdated.
Author Response:
Thank you for the valuable comments and we extend our sincerest gratitude to reviewer. The literature cited in our original letter is more outdated, and we have introduced state-of-the-art methods in the introduction section of the revised version to modify the literature review. (Revised edition: P2, Line 65-88)
All in all, thank you very much for your reconsidering our revised manuscript for potential publication in Sensors. I'm looking forward to hearing from you soon. Correspondence should be addressed to Wanjing Yan or Weifeng Ma at the following address.

Reviewer 3 Report
Comments and Suggestions for Authors
While an interesting topic, there are several points which require clarification before publication:
1) The author's work focuses on using a cloth filter to remove excess ground points in order to rebalance the training data. However, it is unclear if the observed effects are a result of the chosen algorithm or just the rebalancing of the data. Because the ground points are already identified, it would be more compelling to compare the results of the cloth filter against a random reduction in ground points used in the training of the ML classification algorithms.
2) ML hyperparameter tuning isn't explicitly mentioned, and one has to assume some sort of default hyperparameter values were used. This would explain the poor performance of the SVM model, which is very sensitive to hyper-parameter values. As such, it is unclear how valid the presented results are: better tuned models could have very different performance.
3) Model validation isn't mentioned in the manuscript. As such, one has to assume the presented results are based on in-sample predictions. If that's the case, the generalizability of the findings are unclear.
4) Adaboost is an optimization algorithm, often used in Neural Networks. When the authors mention Adaboost, are they referring to a neural network? Or perhaps some other application of this optimization algorithm?
Comments on the Quality of English Language
The manuscript contains significant grammatical errors which create barriers to understanding it's content. For example, in the Abstract:
Line 14: "mining" is the wrong term. Perhaps "identifying"?
Line 15: Are there a massive number of point clouds? Or are there a massive number of points in point clouds?
Line 18-19: The phrase "the ground point with the largest number" is non-sensical
Line 19: What is 'mature' cloth?
Line 22: Missing article in front of transmission
There are more similar errors, and editing for clarity is required.
Author Response
Response to Reviewers’ Comments
Remote sensing
Manuscript Number: sensors-3211209
Manuscript Title: Filtering-assisted airborne point cloud semantic segmentation for transmission lines
General Comments: We are extremely grateful to the esteemed editors and reviewers for reviewing my work and providing detailed comments and suggestions for revision. I really appreciate Editor giving me a chance to revise. Thanks a lot. I have revised the original letter according to reviewers, including revising abstract and introduction, increasing validation experiments,enhancing the discussion section,adding explanations, polishing the letter and so on. A revision location is marked in the letter with a notation like (Revised edition: P1, Line 12-36), which means that the revision occurs on page 1, from Line 12 in the left side to the Line 36 in the right side of the document .
A list of changes and responses to reviewers are as follows. A copy of the fully revised manuscript that has all the changes highlighted in red color(Revised edition_highlight in red), along with grammatical corrections(Revised edition_remarked) is uploaded as an attachment to the email as well. Appreciate for your consideration.
Reviewer #3:
While an interesting topic, there are several points which require clarification before publication:
1)The author's work focuses on using a cloth filter to remove excess ground points in order to rebalance the training data. However, it is unclear if the observed effects are a result of the chosen algorithm or just the rebalancing of the data. Because the ground points are already identified, it would be more compelling to compare the results of the cloth filter against a random reduction in ground points used in the training of the ML classification algorithms.
Author Response:
Thank you for pointing out these problems.Our initial letter did not clearly explain whether the observed effect was a result of the selected algorithm or was due to a rebalancing of the data, while I also think that your advice is very consistent with our ideas. We added a contrast experiment with different filtering levels based on the original experimental data to enhance the persuasion of this algorithm,As shown in Figure 3. (Revised edition: P6, Line 235)
At the same time, we added three sets of comparison verification data to validate the semantic segmentation performance of the method presented in this paper under different ground point filtering effects and varying topographic conditions in transmission line scenarios. Additionally, considering factors such as terrain undulation degree, number of point clouds, class imbalance, and number of categories, we will conduct a correlation analysis between the results from different classification algorithms and these influencing factors. This will enhance the persuasiveness of our method and also allow us to compare and analyze the applicability of different classification methods. Thanks for the comments.(Revised edition: P11, Line 368-498).
- ML hyperparameter tuning isn't explicitly mentioned, and one has to assume some sort of default hyperparameter values were used. This would explain the poor performance of the SVM model, which is very sensitive to hyper-parameter values. As such, it is unclear how valid the presented results are: better tuned models could have very different performance.
Author Response:
Firstly,I would like to express my gratitude for the useful comments on the letter. We considered the impact of parameter settings of machine learning classifiers on the semantic segmentation of point clouds based on power line scenarios. Due to the low classification accuracy under the default parameters of Adaboost and SVM algorithms, we optimized the parameters of these two algorithms. While the other three algorithms adopt the default parameters. The relevant introduction has been described in the revision of the manuscript.(Revised edition: P7, Line 276-291)
Secondly,Thank you for your valuable advice, and I strongly agree with it. We recognize that it is unreasonable to compare the performance of classifiers without unified parameter tuning. Therefore, when achieving good classification performance (either through parameter tuning or using default parameters), keeping the parameter settings of individual classifiers consistent before and after filter assistance is more meaningful for investigating the impact of filter assistance on the algorithms.
- Model validation isn't mentioned in the manuscript. As such, one has to assume the presented results are based on in-sample predictions. If that's the case, the generalizability of the findings are unclear.
Author Response:
I would like to express my gratitude for the useful comments on the letter. In view of the lack of model validation in the manuscript and the lack of universality of the experimental results, we selected two sets of validation data, data2 and data3, for comparative experiments to verify the universality of the experimental results. The insightful and helpful suggestions are greatly appreciated. As shown in Figure 1(Revised edition: P3, Line 151).
At the same time, the relevant experimental results are also displayed and discussed in the section of "3.3." through the combination of text and text.(Revised edition: P11, Line 368-498)
- Adaboost is an optimization algorithm, often used in Neural Networks. When the authors mention Adaboost, are they referring to a neural network? Or perhaps some other application of this optimization algorithm?
Author Response: I would like to express my gratitude for the useful comments on the letter. The Adaboost algorithm used in this paper uses simple feature-based weak classifiers (Decision Tree) to improve the performance of weak classifiers (non-neural network correlation).
Comments on the Quality of English Language:
The manuscript contains significant grammatical errors which create barriers to understanding it's content. For example, in the Abstract:
Line 14: "mining" is the wrong term. Perhaps "identifying"?
Author Response: We sincerely thank the reviewer for careful reading. As suggested by the reviewer, we have corrected the“mining”into“identifying”.
Line 15: Are there a massive number of point clouds? Or are there a massive number of points in point clouds?
Author Response: We were really sorry for our careless mistakes. Thank you for your reminder. What I want to say is: “the number of point clouds is huge”. We have made changes to this fuzzy expression.
Line 18-19: The phrase "the ground point with the largest number" is non-sensical
Author Response: We sincerely thank the reviewer for careful reading. Thank you for your reminder. What I want to say is: “the number of point clouds is huge”. We have made changes to this fuzzy expression.
Line 19: What is 'mature' cloth?
Author Response: We sincerely thank the reviewer for careful reading. Thank you for your reminder. What I want to say is: “well-developed”. We have made changes to this wrong expression.
Line 22: Missing article in front of transmission
Author Response: We were really sorry for our careless mistakes. Thank you for your reminder. We added the prefix "the" and corrected it in the revised manuscript.
There are more similar errors, and editing for clarity is required.
Author Response: Thank you for your reminder. We tried our best to improve the manuscript and made some changes to the manuscript. These changes will not influence the content and framework of the paper. And here we did not list the changes but marked in red in the revised paper. We appreciate for Editors/Reviewers’warm work earnestly and hope that the correction will meet with approval.

Round 2
Reviewer 2 Report
Comments and Suggestions for Authors
Authors have addressed all my concerns
Reviewer 3 Report
Comments and Suggestions for Authors
Overall, I believe that the current version of manuscript is much improved over the original. Thank you for taking my previous comments and suggestions to heart.
Comments on the Quality of English Language
There remain several minor errors in manuscript, but these errors rarely obscure the meaning of text. For example:
Line 16: Grammar
Line 66: Typo